# Glycosaminoglycans from the Starfish *Lethasterias fusca*: Structures and Influence on Hematopoiesis

**DOI:** 10.3390/md21040205

**Published:** 2023-03-24

**Authors:** Maria I. Bilan, Natalia Yu. Anisimova, Alexandra I. Tokatly, Sofya P. Nikogosova, Dmitriy Z. Vinnitskiy, Nadezhda E. Ustyuzhanina, Andrey S. Dmitrenok, Evgenia A. Tsvetkova, Mikhail V. Kiselevskiy, Nikolay E. Nifantiev, Anatolii I. Usov

**Affiliations:** 1N.D. Zelinsky Institute of Organic Chemistry, Russian Academy of Sciences, Leninsky Prospect 47, Moscow 119991, Russia; 2FSBI N.N. Blokhin National Medical Research Center of Oncology, Kashirskoye sh. 24, Moscow 115458, Russia

**Keywords:** dermatan sulfate, heparinoid, starfish, *Lethasterias fusca*, synthetic sulfated iduronosides, hematopoiesis

## Abstract

Crude anionic polysaccharides extracted from the Pacific starfish *Lethasterias fusca* were purified by anion-exchange chromatography. The main fraction **LF,** having MW 14.5 kDa and dispersity 1.28 (data of gel-permeation chromatography), was solvolytically desulfated and giving rise to preparation **LF-deS** with a structure of dermatan core [→3)-β-d-GalNAc-(1→4)-α-l-IdoA-(1→]_n_, which was identified according to NMR spectroscopy data. Analysis of the NMR spectra of the parent fraction **LF** led to identification of the main component as dermatan sulfate **LF-Derm** →3)-β-d-GalNAc4R-(1→4)-α-l-IdoA2R3S-(1→ (where R was SO_3_ or H), bearing sulfate groups at O-3 or both at O-2 and O-3 of α-l-iduronic acid, as well as at O-4 of some N-acetyl-d-galactosamine residues. The minor signals in NMR spectra of **LF** were assigned as resonances of heparinoid **LF-Hep** composed of the fragments →4)-α-d-GlcNS3S6S-(1→4)-α-l-IdoA2S3S-(1→. The 3-O-sulfated and 2,3-di-O-sulfated iduronic acid residues are very unusual for natural glycosaminoglycans, and further studies are needed to elucidate their possible specific influence on the biological activity of the corresponding polysaccharides. To confirm the presence of these units in **LF-Derm** and **LF-Hep**, a series of variously sulfated model 3-aminopropyl iduronosides were synthesized and their NMR spectra were compared with those of the polysaccharides. Preparations **LF** and **LF-deS** were studied as stimulators of hematopoiesis in vitro. Surprisingly, it was found that both preparations were active in these tests, and hence, the high level of sulfation is not necessary for hematopoiesis stimulation in this particular case.

## 1. Introduction

Marine invertebrates belonging to the classes Asteroidea (sea stars, starfishes) and Holothuroidea (sea cucumbers) are known for their peculiar carbohydrate metabolism [1,2,3]. They produce unique sialoglycolipids (gangliosides) [4,5] and complex glycosides containing steroidal or triterpene aglycons and mono- or oligosaccharide moieties, which are usually sulfated. These glycosides possess various biological activities, such as cytotoxic, hemolytic, antibacterial, anti-inflammatory, antitumor and cancer-preventing effects, and are currently being intensely investigated in order to find potential applications in drug development for human and veterinary medicine [6,7]. The body walls of holothuria contain two types of specific sulfated fucose-rich polysaccharides: sulfated fucans (resembling the fucoidans of brown algae) and fucosylated chondroitin sulfates (related to the chondroitin sulfates of vertebrates) [8,9,10,11]. These polysaccharides were shown to be very promising biologically active polymers having various preventive or therapeutic effects on medical conditions [12,13,14,15].

Meanwhile, data on the chemical structures of the corresponding polysaccharides isolated from starfishes are very scarce. A very unusual polysaccharide (composed of pentasaccharide repeating units containing xylose, galactose, fucose and sulfate) was found as to be the acrosome reaction-inducing substance in the starfish *Asterias amurensis* [16], but similar components in the egg jelly coat of other species were apparently not investigated. Crude polysaccharides obtained by extraction of the body walls were mainly characterized as biologically active preparations, but the chemical structures of the polysaccharides were not elucidated. For example, it was shown that polysaccharide samples obtained from *Asterias rollestoni* have neuroprotective and immunostimulating effects [17,18], a polysaccharide from *Asterina pacifica* possess chemopreventive activity against breast cancer and HT-29 human colon adenocarcinoma cells [19,20,21,22], and a polysaccharide from the brittle star *Ophiocoma erinaceus* was found to promote apoptosis [23]. A rather unusual branched α-glucan was isolated recently from the brittle star *Trichaster palmiferus* [24]. According to preliminary evidence [25], structural sulfated glycosaminoglycans may have widespread occurrence in the body walls of starfishes. This suggestion was confirmed by the detection of highly sulfated chondroitin sulfates/dermatan sulfates in several brittle stars [26,27]. In addition, the presence of oversulfated dermatan sulfate and heparinoid in the body walls of the starfish *Lysastrosoma anthosticta* was described in our previous work [28].

The present paper is devoted to the isolation and characterization of sulfated polysaccharides from the starfish *Lethasterias fusca.* This species was used previously as the source of gangliosides [29], asterosaponins, polyhydroxysteroids, as well as their sulfates and glycosides [30,31]. Many representatives of these unique metabolites have been isolated and structurally characterized [32]. Moreover, the distribution of polar steroids and glycoconjugates in different organs of the starfish was analyzed in order to understand the biological functions of these compounds [33], but polysaccharide composition was not investigated.

We have isolated a highly sulfated polysaccharide preparation **LF** from the body walls of *L. fusca*. Desulfated derivative **LF-deS** was prepared from **LF** by solvolytic procedure. The structures of **LF** and **LF-deS** were elucidated using chemical analysis and NMR spectroscopy data. It was shown that **LF** is a mixture of two components, highly sulfated dermatan sulfate **LF-Derm** and heparinoid **LF-Hep** in a ratio of about 4:1, with both polysaccharides containing unusual 2,3-di-O-sulfated α-l-iduronic acid residues. In the structure of **LF-Derm**, another unusual structural fragment, namely 3-O-sulfated α-l-iduronic acid residue, was observed. Polysaccharide preparations **LF** and **LF-deS** were studied as stimulators of hematopoiesis.

## 2. Results and Discussion

The body walls of the starfish *Lethasterias fusca* were extracted in the presence of papain [34] to obtain a crude preparation of sulfated polysaccharides **LF-SP**. Then the polysaccharide mixture was fractionated by anion-exchange chromatography on DEAE-Sephacel column. The main fraction **LF** (yield 40.7%) was obtained by elution with 1.0 M NaCl. According to the composition of **LF** (high levels of galactosamine, uronic acid and sulfate, moderate amount of glucosamine, see Section 3.2 and Appendix A), the preparation might be preliminarily regarded as a sulphated polysaccharide belonging to CS-DS group with possible admixture with other GAGs.

The NMR spectra of preparation **LF** contain several signals with different intensity in the anomeric region (Figure 1). Desulfation of **LF** under solvolytic conditions led to product **LF-deS** giving NMR spectra which were easier to interpret. Two main signals were observed in the anomeric region of its ^13^C NMR spectrum (Figure 1). Application of the 2D NMR techniques (COSY, ROESY and HSQC, see the HSQC spectrum in Appendix A) allowed us to assign the signals of the spin systems of units **A** and **B** (Figure 2) both in the ^1^H and ^13^C NMR spectra (Table 1). Comparing these data to those for the desulfated derivative **LA-F1-DS**, obtained previously by modification of the polysaccharide from the starfish *Lysastrosoma anthosticta* [28], revealed the identity of these polysaccharides. It was shown that polysaccharide **LF-deS**, similarly to **LA-F1-DS**, was dermatan with the regular structure [→3)-β-d-GalNAc-(1→4)-α-l-IdoA-(1→]_n_ (Figure 2, the absolute configurations of monosaccharide residues were ascribed by analogy with other natural dermatan sulfates).

Then, the detailed assignment of the signals in the NMR spectra of the parent preparation **LF** was performed using the 2D COSY, TOCSY, ROESY, HSQC experiments (Figure 3, Appendix A, Table 1). It was found that the main component of **LF** was dermatan sulfate bearing sulfate groups at positions 2 and 3 of α-l-iduronic acid (unit **C**), as well as at O-4 of most parts of N-acetyl-d-galactosamine residues (unit **E**). A similar polysaccharide **LA-Derm** was determined previously in the starfish *L. anthosticta* [28]. These structural fragments were found earlier in several oversulfated dermatans isolated from ascidians [35,36] and in a polysaccharide prepared by chemical sulfation of pig intestine dermatan sulfate [37]. We observed a good coincidence in the NMR data of our sample and these oversulfated polysaccharides described previously.

In addition to the IdoA2S3S unit, there was an IdoA3S fragment (unit **D**) in preparation **LF**. The signals of the anomeric H-1 (*δ* 4.97 ppm) and C-1 (*δ* 103.8 ppm) of this unit differed significantly from those of IdoA2S3S (Figure 3, Table 1). The presence of a sulfate group at O-3 was confirmed by the downfield chemical shift of the respective H-3 signal. Attachment of IdoA3S to O-3 of GalNAc unit was confirmed by the presence of the correlation peak H1(**D**)-H3(**E**,**F**) in the ROESY spectrum (Appendix A). Therefore, the dermatan sulfate **LF-Derm** from *L. fusca* can be described by the formula [→3)-β-d-GalNAc4R-(1→4)-α-l-IdoA2R3S-(1→]_n_ (where R was SO_3_^–^ or H) and should be considered as a more complex polysaccharide than that from *L. anthosticta*.

Dermatan sulfate was not the only component of preparation **LF**. The correlation H2-C2 (*δ* 3.47 ppm/57.9 ppm) in the HSQC spectrum (Figure 3B) indicated the presence of glucosamine residue sulfated at N-2 GlcNS (unit **H**) [38]. This unit was also sulfated at O-3 and at O-6 (Table 1). Unit IdoA2S3S (**G**) was found to be connected to O-4 of GlcNS (**H**) (see ROESY spectrum, Appendix A). Therefore, the minor component of **LF** should be considered as heparinoid **LF-Hep** described by the formula [→4)-α-d-GlcNS3S6S-(1→4)-α-l-IdoA2S3S-(1→]_n_. A similar structural fragment was found in an oversulfated polysaccharide obtained by chemical sulfation of heparin [39]. Comparing the chemical shift values of the corresponding signals showed a good coincidence. Nevertheless, there are inevitable differences in the data for polysaccharides, since **LF-Hep** contains the IdoA2S3S residue at O-4 of unit **H**, whereas in the oversulfated heparin, this position is taken up by a sulfated β-glucuronic acid substituent. Previously, a similar heparinoid was determined in the starfish *L. anthosticta* in significant quantities [28]. The ratio of GalNAc and GlcNS residues in **LF** calculated as a ratio between the integral intensities of the respective H2-C2 cross peaks in the HSQC NMR spectrum was found to be 4:1, which practically coincides with the ratio between galactosamine and glucosamine determined by chemical monosaccharide analysis (Appendix A).

The monosaccharide residues IdoA3S and IdoA2S3S are unusual for natural polysaccharides. Model iduronic acid aminopropyl glycosides **7–11** (Figure 1) with different sulfation patterns were synthesized to aid the detailed analysis of the NMR spectra of **LF**. All model compounds were obtained from fully protected intermediate **6**. Readily available thioglycoside **1** [40] was synthesized from glucose pentaacetate, subjected to acetyl cleavage, 4,6-O-*p*-methoxybenzylidene protection and DBTO-promoted regioselective introduction of 2-methylnaphthyl at O-3. Subsequent removal of the acetal protection yielded 3-*O*-protected compound **2**. C-6-oxidation using the TEMPO/BAIB system resulted in glucuronic thioglycoside **3**, which was then subjected to direct epimerization under basic conditions [41] to give iduronic compound **4** in a yield of 50%. Selective 2-O-benzoylation using DBTO and silylation with tert-butyldimethylsilyl trifluoromethanesulfonate in the presence of triethylamine gave fully protected thioglycoside **5**, which was then oxidized by *m*-chloroperbenzoic acid to yield a mixture of diastereomeric sulfoxide glycosyl donors. Those were used in a Tf_2_O-promoted glycosylation reaction with 3-azido-1-propanol as acceptor to obtain compound **6**, which was then subjected to selective deprotection and sulfation, followed by final deprotection, to give the desired glycosides **7–11** (Figure 1).

Analysis of the NMR spectra of model compounds **7–11** (see Appendix A) revealed that sulfation led to significant (0.6–1.5 ppm) downfield shift of the signals of the respective protons and in some cases of the respective carbons (compounds **8** and **9**, Table 1). A similar effect was observed in polysaccharides **LF-Derm** and **LF-Hep** for the units IdoA2S3S (**C**,**G**) and IdoA3S (**D**) in comparison to non-sulfated IdoA (unit **A**). Additionally, 2-O-sulfation influences the signals of the anomeric protons (downfield shift on 0.25–0.30 ppm) and carbons (upfield shift on 1.7–2.1 ppm) (see compounds **8** and **11**). This effect was found for unit IdoA2S3S (**C**), compare *δ* 5.20 ppm for H-1 and 101.8 ppm for C-1 (units **C**) to 4.94 ppm for H-1 and 105.0 for C-1 (unit **A**). Interestingly 3-O-sulfation also slightly influences the value of anomeric proton chemical shift (see compound **9**). Additional introduction of a bulky substituent at neighbor O-4 (see compound **10**) increases the downfield shift of H-1 up to 0.1 ppm. A similar observation was made for unit IdoA3S (**D**) in the polysaccharide **LF-Derm** (compare *δ* 4.97 ppm for H-1 of units **D** to 4.94 ppm for H-1 of unit **A**).

Notably the presence of IdoA3S units in a glycosaminoglycan molecule was observed for the first time. As this manuscript was being prepared for publication, a paper appeared [42] describing IdoA3S residues found in the carbohydrate moiety of some unusual glycoproteins isolated from the halophilic archaeon *Halobacterium salinarum.* The assignments of the IdoA3S signals in the NMR spectra given by Notaro et al. [42] slightly differ from our data, and this inconsistency may be explained by the influence of the varying monosaccharide residues surrounding IdoA3S in the two different biopolymers.

For the preliminary assessment of the molecular weight of preparation **LF**, gel electrophoresis was performed using heparin (Sigma-Aldrich, St. Louis, MO, USA) and enoxaparin (Clexan^®^, Sanofi, Paris, France) with defined MW as a standard (Figure 4). Based on mobility of samples, it could be concluded that MW value of **LF** was quite similar to that of heparin (15 kDa). Further detailed MW determination was performed using gel chromatography. As a result, almost equal values of MW were found for heparin and preparation **LF** (14.5 kDa, dispersity 1.28, Appendix A).

Sulfated polysaccharides isolated from marine plants and animals, such as fucoidans and fucosylated chondroitin sulfates, were shown to stimulate hematopoiesis in mice [43,44,45,46,47]. This type of activity was found to be connected with the ability of sulfated polysaccharides to stimulate the adhesion of bone marrow cells and their further proliferation [28]. Previously, we have shown that fucoidans and fucosylated chondroitin sulfates demonstrated significant activity in concentrations 10–50 µg/mL [43,47]. As we did not know the level of this type of activity for glycosaminoglycans from the starfishes, we have studied the samples from *L. fusca* in the concentration 50 µg/mL, and additionally, in a concentration of 250 µg/mL, which was five times higher. 

To study the influence of glycosaminoglycans from the starfish *L. fusca* on hematopoiesis, samples of **LF** and **LF-deS** were incubated with bone marrow cells for 7 days ex vivo. Recombinant granulocyte colony-stimulating factor (**rG-CSF**) was used as a positive control. The parameter Cell Index (CI) was measured using the Agilent xCELLigence real-time cell analysis multiple plates system (Figure 5). The value of CI is correlated with the ability of the cells to adhere and consequently proliferate. It was found that preparation **LF** in a concentration of 50 µg/mL led to an increase in CI during the first day, but further on a significant decrease in this value was observed, which could be connected with the induction of apoptosis or some toxic effects of **LF**. It was confirmed by the experiment with a five times higher concentration of **LF** (250 µg/mL), where almost no attached cells were detected. On the contrary, preparation **LF-deS** in both tested concentrations led to an increase in CI. The stimulation curve profiles for **LF-deS** were quite similar to that of **rG-CSF** in lower concentration, indicating the ability of **LF-deS** to stimulate cell adhesion and proliferation. Interestingly, a higher concentration of **rG-CSF** also led to a slower increase in CI.

## 3. Materials and Methods

### 3.1. General Methods

Acid hydrolysis of polysaccharide samples followed by preparation of alditol acetates was described previously [28,48,49]. Gas-liquid chromatography was performed using Agilent 8860 GC System equipped with flame-ionization detector and HP-5 chromatographic column 30 m × 0.320 mm × 0.25 μm working at 160–290 °C with nitrogen as carrier gas (Appendix A). Quantitative determination of monosaccharides with *myo*-inositol acetate as internal standard was carried out using OpenLab CDS 2 program. Estimations of sulfate [50], uronic aids [51] and proteins [52] was accomplished using Ultrospec 4050 spectrophotometer (LKB Biochrom, Bromma, Sweden).

The NMR spectra were recorded using the facilities of Zelinsky Institute Shared Center. Preparation of polysaccharide solutions, as well as conditions used to record 1D and 2D proton and carbon-13 COSY, TOCSY, ROESY and HSQC spectra were described earlier [28,53,54]. For synthetic monosaccharide derivatives chemical shifts are reported in ppm referenced either to the solvent residual peaks as standard for chloroform (δ 7.26 ^1^H, δ 77.16 ^13^C) or methanol as internal standard for D_2_O (δ 3.34 ^1^H, δ 49.50 for ^13^C).

Optical rotations were measured using a JASCO P-2000 polarimeter at ambient temperature (22–25 °C).

High-resolution mass spectra (HR MS) were measured on a Bruker micrOTOF II instrument using electrospray ionization (ESI). The measurements were performed in a positive ion mode (interface capillary voltage −4500 V) or in a negative ion mode (3200 V); mass range from *m/z* 50 to *m/z* 3000 Da; external or internal calibration was made with Electrospray Calibrant Solution (Fluka). A syringe injection was used for solutions in a mixture of acetonitrile and water (50:50 *v/v*, flow rate 3 μL/min). Nitrogen was applied as a dry gas; interface temperature was set at 180 °C.

Commercial chemicals for the synthesis of model monosaccharide derivatives were used without purification unless noted. All solvents were distilled and dried if necessary, according to standard procedures (DCM, MeOH) or purchased as dry (DMF, THF, CH_3_CN, Sigma-Aldrich). All reactions involving air- or moisture-sensitive reagents were carried out using dry solvents under Ar atmosphere. Thin-layer chromatography (TLC) was carried out on aluminum sheets coated with silica gel 60 F254 (Merck, Rahway, NJ, USA). TLC plates were inspected by UV light (λ = 254 nm) and developed by treatment with a mixture of 15% H_3_PO_4_ and orcinol (1.8 g/L) in EtOH/H_2_O (95:5, *v/v*) followed by heating. Silica gel column chromatography was performed with Silica Gel 60 (40–63 μm, E. Merck, Darmstadt, Germany). Solvents for column chromatography and thin layer chromatography (TLC) are listed in volume-to-volume ratios. Gel-filtration of the synthetic compounds was performed on a Sephadex G-15 column (400 × 16 mm) by elution with water at a flow rate of 0.5 mL/min, or a Toyopearl HW-40S column (400 × 16 mm) eluted with a 0.1 M solution of AcOH at a flow rate of 0.5 mL/min.

Other experimental conditions were described in detail previously [53,54]. Heparin sodium salt was purchased from Sigma-Aldrich (H5515). Molecular weight of polysaccharides was determined using gel chromatography on an analytical TSK 2000 SW_XL_ column (Toyo Soda, Tokyo, Japan, 7.5 × 300 mm) calibrated using pullulans (Fluka, Buchs, Switzerland) [55] at a flow rate of 0.75 mL/min by elution with 1 M NaCl in PBS with refractometer detection at 35 °C.

### 3.2. Isolation of Sulfated Polysaccharides

The starfish *Lethasterias fusca* (Djakonov, 1931) was collected in the littoral zone of the Peter the Great Bay (the Sea of Japan) and identified by the late Prof. V.S Levin (Pacific Institute of Bioorganic Chemistry, FEB RAS, Vladivostok, Russia). After removing the viscera, the body walls were treated several times with methanol and stored under methanol in a refrigerator. According to the conventional procedure [30], dried and minced biomass (227 g) was suspended in 300 mL of 0.1 M sodium acetate buffer (pH 6.0), containing papain (1 g), EDTA (0.4 g), and l-cysteine hydrochloride (0.2 g), and incubated at 45–50 °C for 24 h with occasional agitation. After centrifugation, an aqueous hexadecyl-trimethylammonium bromide solution (10%, 25 mL) was added to the supernatant, and the mixture was allowed to sit overnight. The resulting precipitate was isolated by centrifugation and washed successively with water and ethanol. Then, it was stirred with 20% ethanolic NaI solution (5 × 40 mL) for 2–3 days, washed with ethanol, dissolved in water and lyophilized to give the crude polysaccharide preparation **LF-SP**, yield 0.19%, composition: sulfate, 23.9%; uronic acids, 9.6%; galactosamine, 7.2%; glucosamine, 1.8%; galactose, 2.2%.

An aqueous solution of **LF-SP** (410 mg) was placed on a column (3 × 10 cm) with DEAE-Sephacel in Cl^−^-form and eluted with water followed by NaCl solution of increasing concentration (0.5, 1.0 and 1.5 M) each time until the eluate no longer gave a positive reaction for carbohydrates [56]. The main 1.0 M fraction was desalted on Sephadex G-15 column and lyophilized, giving rise to preparation **LF**, yield 150 mg, composition: sulfate, 36.7%; uronic acids, 28.3%; galactosamine, 12.2%; glucosamine, 3.3%; galactose, 2.7%; protein, 4.6%.

### 3.3. Chemical Modification of Polysaccharide LF

Solvolytic desulfation of **LF** (as pyridinium salt) was carried out as described earlier [48,49]. Briefly, a solution of **LF** (100 mg) in a DMSO-MeOH mixture (9:1, 10 mL) was heated at 80 °C for 5 h, diluted with water, made slightly alkaline (pH about 8) by addition of 3 M aqueous NaOH (0.2 mL); the precipitate formed was separated by centrifugation, dissolved in water and chromatographed on a column containing Sephadex G-15. A high-molecular-weight fraction was lyophilized to afford desulfated preparation **LF-deS** in a yield of 43 mg, composition: sulfate, 1.6%; uronic acids, 50.8%; galactosamine, 21.2%; glucosamine, 6.3%; galactose, 3.0%; protein, 8.2%.

### 3.4. Polyacrylamide Gel Electrophoresis (PAGE)

The polysaccharides **LF**, heparin and enoxaparin (15 μg) were investigated using PAGE under conditions described earlier [28].

### 3.5. Model Monosaccharide Synthesis

Detailed synthetic procedures for the preparation of the model monosaccharides are presented in the Appendix A.

### 3.6. Cell Model

Bone marrow cells (BM cells) were isolated from the femoral bone of healthy Balb/c mice (male, weight 19 ± 1 g). BM cells were suspended in the complete growth medium based on Dulbecco’s modified Eagle’s medium (DMEM) (Sigma-Aldrich, St. Louis, MO, USA), supplemented with 10% fetal bovine serum (FBS; HyClon, Thermo Fisher Scientific, Waltham, Massachusetts, USA), 1% penicillin/streptomycin (PanEco, Moscow, Russia) and 4 mM L-glutamine (PanEco, Moscow, Russia) at 37 °C in atmosphere with 5% CO_2_ to a concentration of 500,000 cells/mL. To study cell activity, the suspension of BM cells (150 μL) was placed in E-plates 16 (ACEA Biosciences, San Diego, CA, USA). Solutions of **LF**, **LF-deS**, and **rG-CSF** in isotonic sodium chloride solution (50 μL) were added to the cells to a concentration of 50 µg/mL or 250 µg/mL for polysaccharides and 0.15 nmol/mL or 0.75 nmol/mL for **rG-CSF**. BM cells supplemented with 50 μL isotonic sodium chloride solution were used as an intact Control. The results were examined daily by assessing the change in the Cell Index in comparison with the Control, which was measured using the Agilent xCELLigence real-time cell analysis multiple plates system (xCELLigence RTCA DP, ACEA Biosciences, San Diego, CA, USA) during incubation for 7 days at 37 °C in atmosphere with 5% CO_2_. A fresh portion of the growth medium (50 μL) was added every two days in each well.

### 3.7. Statistical Analysis

Determinations of the biological activity mentioned in Section 3.6 were performed in quadruplicate (n = 4). The results are presented as Mean ± SD. Statistical significance was determined with Student’s *t* test. *p* values less than 0.05 were considered as significant.

## 4. Conclusions

This work represents the first investigation of glycosaminoglycans from *L. fusca.* Crude sulfated polysaccharides **LF-SP** were isolated from the defatted body walls of the starfish by a conventional procedure and purified further using anion-exchange chromatography to obtain the main heavily sulfated fraction **LF**. Both galactosamine and glucosamine were found in hydrolysate of this preparation, suggesting that it contained at least two different glycosaminoglycans. Their separation was not possible evidently due to their close similarity in molecular weight and sulfate content. Solvolytic treatment of **LF** gave rise to desulfated preparation **LF-deS**.

According to NMR spectroscopy data, the main component of **LF-deS** was the well-known dermatan [→3)-β-d-GalNAc-(1→4)-α-l-IdoA-(1→]_n_. Interpretation of the complex NMR spectra of **LF** was rather difficult and required synthesis of several model compounds. These models included 3-aminopropyl glycoside of α-L-idopyranosyluronic acid and four of its sulfated derivatives differed in number and position of sulfate groups. The signals in the NMR spectra of these models were carefully assigned and gathered together in Table 1, giving valuable information about the influence of sulfate position on chemical shifts of resonances in NMR spectra. Taking into account this evidence, the spectra of **LF** were interpreted. These spectra corresponded to a mixture of two components, an oversulfated dermatan sulfate **LF-Derm** →3)-β-d-GalNAc4R-(1→4)-α-l-IdoA2R3S-(1→ (where R was SO_3_^−^ or H) and a heparinoid **LF-Hep** composed of the fragments →4)-α-d-GlcNS3S6S-(1→4)-α-l-IdoA2S3S-(1→. The ratio between **LF-Derm** and **LF-Hep** was calculated as about 4:1. It should be emphasized that α-l-IdoA3S is uncommon for known dermatan sulfates, and α-l-IdoA2S3S is also a rare component of dermatan sulfates and heparinoids.

In order to determine the biological activity of **LF** and **LF-deS**, both preparations were tested as possible stimulators of hematopoiesis. It was shown that **LF** in low concentration promoted the adhesion and further proliferation of bone marrow cells at the beginning of experiment, but further on, a significant decrease in this effect was observed, possibly linked with the induction of apoptosis or toxic effects of **LF**. It is very interesting that **LF-deS** demonstrated similar action, and hence, this influence on hematopoiesis does not depend directly on the sulfation of polysaccharides. Further investigations are highly desirable in order to explain these unusual properties and determine the structure-activity relationship of these biologically active compounds, as well as elucidating the actual mechanisms behind their activity. It is very probable that some other types of biological activity, characteristic for sulfated glycosaminoglycans, may be found for LF and LF-deS.

## Data Availability

Data sharing not applicable.

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
