# Peer review of "Glycosaminoglycans from the Starfish Lethasterias fusca: Structures and Influence on Hematopoiesis"

_marinedrugs, 2023, doi:10.3390/md21040205_

Round 1
Reviewer 1 Report
The manuscript describes isolation and structural analysis of the sulfated glucosaminoglycans from a starfish L. fusca. Isolated compounds were tested in vitro for stimulation of hematopoiesis. For comparison some fragments of the natural polysaccharides were synthesized. The work is well planned, performed and described.
Author Response
Reviewer 1
Comments and Suggestions for Authors
The manuscript describes isolation and structural analysis of the sulfated glucosaminoglycans from a starfish L. fusca. Isolated compounds were tested in vitro for stimulation of hematopoiesis. For comparison some fragments of the natural polysaccharides were synthesized. The work is well planned, performed and described.
Authors: We are greatly indebted to the Reviewer for positive valuation of the manuscript.
Reviewer 2 Report
Title:
The title is concise, informative, and likely to attract the interest of those in the field of hematology and glycosaminoglycan research.
Abstract:
- The structure of the main polysaccharide fraction LF should be described in greater detail, including the degree of polymerization.
- The significance of the unusual sulfation pattern of the iduronic acid residues should be discussed.
- The methods and results of the hematopoiesis stimulation assays should be described in more detail.
- The abstract could benefit from a brief conclusion summarizing the significance of the findings.
Key words:
The keywords appear to be appropriate and related to the Mesh terms
Introduction:
The introduction is clear and well-written, providing background information on the unique carbohydrate metabolism of marine invertebrates, particularly Asteroidea and Holothuroidea. It also highlights the biological activities of their complex glycosides and sulfated fucose-rich polysaccharides, which have potential applications in drug development for human and veterinary medicine. The introduction also sets the stage for the current study, which focuses on the isolation and characterization of sulfated polysaccharides from the starfish Lethasterias fusca. The only comment for correction is that "sialoglycolipids" should be "gangliosides" in line 4.
Results and discussion:
The results and discussion section is clear and sound. The authors have provided a detailed description of their methodology and findings, and have used appropriate figures and tables to illustrate their results. The language used is scientific and technical, but understandable to the target audience. There are no obvious errors or corrections needed in this section.
Materials and methods:
Are complete and clear and I do not have any comment.
Conclusions
Overall, the conclusion section provides a good summary of the research findings and highlights the novelty and significance of the study. However, some minor improvements can be made.
First, it would be helpful to include a brief statement on the implications of the findings. For example, how might these newly discovered glycosaminoglycans from L. fusca be used in biomedical or industrial applications?
Second, the sentence "Further investigations are highly desirable" could be more specific. What aspects of the study require further investigation? Are there any particular experiments that should be conducted to gain a better understanding of the biological activity of LF and LF-deS?
Lastly, it may be beneficial to include a final statement that emphasizes the importance of continuing research in this area. This could encourage other researchers to build on this study and further explore the potential applications of these glycosaminoglycans.
Author Response
Reviewer 2
Authors: We are greatly indebted to the Reviewer 2 for valuable comments and suggestions.
Comments and Suggestions for Authors
Title:
The title is concise, informative, and likely to attract the interest of those in the field of hematology and glycosaminoglycan research.
Abstract:
- The structure of the main polysaccharide fraction LF should be described in greater detail, including the degree of polymerization.
Authors: MW value of fraction LF, determined by GPC, is now mentioned in line 15: The main fraction LF having MW 14.5 kDa and dispersity 1.28 (data of gel-permeation chromatography)…
- The significance of the unusual sulfation pattern of the iduronic acid residues should be discussed.
Authors: The text in lines 23-24 is modified as follows: The 3-O-sulfated and 2,3-di-O-sulfated iduronic acid residues are very unusual for natural glycosaminoglycans, and further studies are needed to elucidate their possible specific influence on the biological activity of the corresponding polysaccharides.
- The methods and results of the hematopoiesis stimulation assays should be described in more detail.
Authors: The methods of hematopoiesis stimulation assays are described in detail in the corresponding Section 3.6. Importance of the results is emphasized shortly according to the next remark of the Reviewer.
- The abstract could benefit from a brief conclusion summarizing the significance of the findings.
Authors: The text added to line 27: Surprisingly it was found that both preparations were active in these tests, and hence, the high level of sulfation is not necessary for hematopoiesis stimulation in this particular case.
Key words:
The keywords appear to be appropriate and related to the Mesh terms
Introduction:
The introduction is clear and well-written, providing background information on the unique carbohydrate metabolism of marine invertebrates, particularly Asteroidea and Holothuroidea. It also highlights the biological activities of their complex glycosides and sulfated fucose-rich polysaccharides, which have potential applications in drug development for human and veterinary medicine. The introduction also sets the stage for the current study, which focuses on the isolation and characterization of sulfated polysaccharides from the starfish Lethasterias fusca. The only comment for correction is that "sialoglycolipids" should be "gangliosides" in line 4.
Authors: The terms "sialoglycolipids" and "gangliosides" are equivalent in meaning, but differ in their origin. The first name reflects chemical nature of substances, whereas the second one is connected with their location in organisms. They both are mentioned in line 34 of the Introduction, and we prefer to retain them in the text.
Results and discussion:
The results and discussion section is clear and sound. The authors have provided a detailed description of their methodology and findings, and have used appropriate figures and tables to illustrate their results. The language used is scientific and technical, but understandable to the target audience. There are no obvious errors or corrections needed in this section.
Materials and methods:
Are complete and clear and I do not have any comment.
Conclusions
Overall, the conclusion section provides a good summary of the research findings and highlights the novelty and significance of the study. However, some minor improvements can be made.
First, it would be helpful to include a brief statement on the implications of the findings. For example, how might these newly discovered glycosaminoglycans from L. fusca be used in biomedical or industrial applications?
Authors: We have made only very preliminary experiments demonstrating biological activity of LF as stimulator of hematopoiesis. The data obtained are quite insufficient for conclusion about use of the preparation in biomedical or industrial applications.
Second, the sentence "Further investigations are highly desirable" could be more specific. What aspects of the study require further investigation? Are there any particular experiments that should be conducted to gain a better understanding of the biological activity of LF and LF-deS?
Authors. It is important to check other types of biological activity, which are usually found for sulfated polysaccharides and the corresponding phrase was added in line 403: It is very probable that some other types of biological activity, characteristic for sulfated glycosaminoglycans, may be found for LF and LF-deS.
Lastly, it may be beneficial to include a final statement that emphasizes the importance of continuing research in this area. This could encourage other researchers to build on this study and further explore the potential applications of these glycosaminoglycans.
Authors. We think that the added sentence corresponds to this remark.
Reviewer 3 Report
Determination of structure-activity relationship within natural polysaccharides is a challenging task of modern chemistry. Structural features, including monosaccharide composition, types of glycoside bonds, presence of charged groups, pattern of sulfation, influence significantly on a type and a level of biological activity.
In the manuscript marinedrugs-2282082 Glycosaminoglycans from the starfish Lethasterias fusca: structures and influence on hematopoiesis. Study of biological activities of the obtained polysaccharides was performed as well.
· Although the stimulators of hematopoiesis in vitro propertie was the main part of the authors' studies, the introduction about the relationship between polysaccharide and hematopoiesis was insufficient. The author should rewrite the relevant section.
· The authors have performed a very poor revision of the pertinent literature, neglecting important data already published regarding structural characteristics of dermatan sulfate and chondroitin sulfate.
· Determination of the molecular weight of the polysaccharides. Application of the gel permission chromatography is required.
· Line 314 : papain (1g) ????? how much Unit ??? with this amount of protease the cos twill be very high ???
· Line 358 : authors studied only two concentration for cell activity 50 and 250 µg/ml. Why ??
· Please add the picture of starfish Lethasterias fusca and the sulfated polysaccharide (to show the color).
· Conclusion must be rerwitten, too long.
Author Response
Reviewer 3
Authors: We are greatly indebted to the Reviewer for valuable comments and suggestions.
Comments and Suggestions for Authors
Determination of structure-activity relationship within natural polysaccharides is a challenging task of modern chemistry. Structural features, including monosaccharide composition, types of glycoside bonds, presence of charged groups, pattern of sulfation, influence significantly on a type and a level of biological activity.
In the manuscript marinedrugs-2282082 Glycosaminoglycans from the starfish Lethasterias fusca: structures and influence on hematopoiesis. Study of biological activities of the obtained polysaccharides was performed as well.
Although the stimulators of hematopoiesis in vitro propertie was the main part of the authors' studies, the introduction about the relationship between polysaccharide and hematopoiesis was insufficient. The author should rewrite the relevant section.
Authors. The main part of the studies is elucidation of the structure of preparation LF. Therefore, data given in Introduction are connected mainly with chemical structures of complex carbohydrates present in echinoderms. We have made only very preliminary experiments demonstrating biological activity of LF as stimulator of hematopoiesis, and hence, describe the problem and cite the corresponding references [28, 43-47] in lines 233-251 of Section 2 “Results and Discussion”.
The authors have performed a very poor revision of the pertinent literature, neglecting important data already published regarding structural characteristics of dermatan sulfate and chondroitin sulfate.
Authors. Our literature review was restricted to references devoted to glycosaminoglycans present in echinoderms.
Determination of the molecular weight of the polysaccharides. Application of the gel permission chromatography is required.
Authors. Application of gel permeation chromatography for determination of molecular weight was mentioned in lines 224-227 of Section 2, in lines 303-305 of Section 3.1, and (after corrections) in the Abstract.
Line 314: papain (1g)????? how much Unit??? with this amount of protease the cos twill be very high???
Authors. Extraction procedure in the presence of papain was suggested many years ago [34] and used many times by different authors. Papain for this procedure is an inexpensive crude powder prepared from papaya latex (for example, P-3375 from Sigma, activity 1.5-10 units/mg solid).
Line 358: authors studied only two concentration for cell activity 50 and 250 µg/ml. Why??
Authors. We have added the explanation in Results "Previously we have shown that fucoidans and fucosylated chondroitin sulfates demonstrated significant activity in concentrations 10-50 µg/mL [43, 47]. As we did not know the level of this type of activity for glycosaminoglycans from the starfishes, we have studied the samples from L. fusca in the concentration 50 µg/mL and additionally in 5 times higher concentration 250 µg/mL".
Please add the picture of starfish Lethasterias fusca and the sulfated polysaccharide (to show the color).
Authors. We have no our own pictures of the starfish Lethasterias fusca. The sulfated polysaccharide looks as a slightly cream-colored powder, but a rather small amount of sample being at our disposal precluded obtaining the picture of good quality.
Conclusion must be rerwitten, too long.
Authors. In our opinion, conclusion cannot be shortened, but it was corrected slightly according to recommendation of Reviewer 2: It is very probable that some other types of biological activity, characteristic for sulfated glycosaminoglycans, may be found for LF and LF-deS.
Round 2
Reviewer 3 Report
The paper can be accepted in its present form